# Soluble Protein Hydrolysate Ameliorates Gastrointestinal Inflammation and Injury in 2,4,6-Trinitrobenzene Sulfonic Acid-Induced Colitis in Mice

**DOI:** 10.3390/biom12091287

**Published:** 2022-09-13

**Authors:** Jingjing Wei, Guozhong Tao, Baohui Xu, Kewei Wang, Junlin Liu, Chih-Hsin Chen, James C. Y. Dunn, Crawford Currie, Bomi Framroze, Karl G. Sylvester

**Affiliations:** 1Department of Surgery, Stanford University School of Medicine, Stanford, CA 94304, USA; 2Department of Pediatrics, Shanxi Medical University, Taiyuan 030001, China; 3Department of Gastrointestinal Surgery, The First Hospital of China Medical University, Shenyang 110001, China; 4Department of General Surgery, The People’s Hospital of Liuyang City, Liuyang 410300, China; 5R&D Department, Hofseth BioCare AS, 6003 Aalesund, Norway

**Keywords:** soluble protein hydrolysate, IBD, TNBS, anti-inflammatory, anti-oxidative

## Abstract

Inflammatory bowel diseases (IBD) are chronic, recurring gastrointestinal diseases that severely impair health and quality of life. Although therapeutic options have significantly expanded in recent years, there is no effective therapy for a complete and permanent cure for IBD. Well tolerated dietary interventions to improve gastrointestinal health in IBD would be a welcome advance especially with anticipated favorable tolerability and affordability. Soluble protein hydrolysate (SPH) is produced by the enzymatic hydrolysis of commercial food industry salmon offcuts (consisting of the head, backbone and skin) and contains a multitude of bioactive peptides including those with anti-oxidant properties. This study aimed to investigate whether SPH ameliorates gastrointestinal injury in 2,4,6-trinitrobenzene sulfonic acid (TNBS)-induced mouse colitis model. Mice were randomly assigned to four groups: Control (no colitis), Colitis, Colitis/CP (with control peptide treatment), and Colitis/SPH (with SPH treatment). Colitis was induced by cutaneous sensitization with 1% TNBS on day −8 followed by 2.5% TNBS enema challenge on day 0. Control peptides and SPH were provided to the mice in the Colitis/CP or Colitis/SPH group respectively by drinking water at the final concentration of 2% *w*/*v* daily from day −10 to day 4. Then, the colon was harvested on day 4 and examined macro- and microscopically. Relevant measures included disease activity index (DAI), colon histology injury, immune cells infiltration, pro- and anti-inflammatory cytokines and anti-oxidative gene expression. It was found that SPH treatment decreased the DAI score and colon tissue injury when compared to the colitis-only and CP groups. The protective mechanisms of SPH were associated with reduced infiltration of CD4^+^ T, CD8^+^ T and B220^+^ B lymphocytes but not macrophages, downregulated pro-inflammatory cytokines (tumor necrosis factor-α and interleukin-6), and upregulated anti-inflammatory cytokines (transforming growth factor-β1 and interleukin-10) in the colon tissue. Moreover, the upregulation of anti-oxidative genes, including ferritin heavy chain 1, heme oxygenase 1, NAD(P)H quinone oxidoreductase 1, and superoxide dismutase 1, in the colons of colitis/SPH group was observed compared with the control peptide treatment group. In conclusion, the protective mechanism of SPH is associated with anti-inflammatory and anti-oxidative effects as demonstrated herein in an established mice model of colitis. Clinical studies with SPH as a potential functional food for the prevention or as an adjuvant therapy in IBD may add an effective and targeted diet-based approach to IBD management in the future.

## 1. Introduction

Inflammatory bowel diseases (IBD) affect more than 1.5 million Americans, with over 70,000 new cases diagnosed annually (https://www.crohnscolitisfoundation.org/sites/default/files/2019-02/Updated%20IBD%20Factbook.pdf (accessed on 4 September 2022)). IBD result from a complex interplay of pathophysiologic factors including genetics, immune system activation, gastrointestinal dysbiosis, and environmental. Given a typical peak age of onset prior to 35, IBD as chronic autoimmune diseases result in chronic and recurrent inflammatory and oxidative injury to the human gastrointestinal tract [1,2,3,4]. Crohn’s disease and ulcerative colitis are the two major types of IBD, sharing many clinical symptoms but having quite different pathological features. Crohn’s disease can affect any part of the gastrointestinal tract from the mouth to the anus, with areas of damage appearing in patches adjacent to areas of healthy tissue and affecting all layers of the gastrointestinal tract. In contrast, ulcerative colitis exclusively affects the large intestine and the rectum, with continuous involved areas of inflammation that are normally limited to the mucosal lining [3].

As the incidence of IBD has risen in recent years in both industrialized and emerging economies and developing countries [5], substantial research investment has provided several novel treatment options. Still, there is no effective therapy for a complete and permanent cure for IBD, and chronic relapse is frequent, necessitating disease management during periods of quiescence and flare. The primary medical treatments for IBD include salicylates, corticosteroids, and immunomodulators in order to reduce inflammation, ease symptoms, and reduce flare intensity. However, with most of these medication classes, their use is associated with unwanted side effects and or the emergence of drug specific resistance. The consequent need for lifelong management in IBD places a significant economic burden on the patient and society [5].

Recently, probiotics and dietary supplements have emerged as potential treatment options with measurable impact on disease severity documented in clinical research reports [6,7]. Protein hydrolysates consist of a mixture of varying proteins and peptides which are formed by the hydrolysis of intact proteins and are commonly used as an alternative protein source in commercial products [8,9]. Protein hydrolysates are also recognized as a potent source of bioactive peptides that have different molecular weights and varying biological effects, with anti-thrombotic, anti-hypertensive, anti-microbial, anti-cancer, anti-oxidative, and immunomodulatory effects having been identified [7,8,9,10]. Soluble protein hydrolysate (SPH) is produced by the enzymatic hydrolysis of salmon offcuts consisting of the head, backbone and skin, and contains a variety of bioactive peptides with reported health benefits including against oxidative stress [11,12,13]. Therefore, the current study was conducted to investigate whether SPH can be used as a functional food to ameliorate gastrointestinal injury in the 2,4,6-trinitrobenzene sulfonic acid (TNBS)-induced IBD mouse model and to explore the protective mechanism of observed effects.

## 2. Materials and Methods

2.1. Experimental Design for TNBS-Induced Colitis model [14]: Experiments were conducted on 8-week-old female BALB/c mice (The Jackson Laboratory, Bar Harbor, ME, USA) that were bred in an animal facility at Stanford University. According to the experimental procedure as shown in Figure 1, 31 mice were randomly divided into 4 groups: Control (no colitis, n = 7), Colitis (n = 8), Colitis/CP (with control peptide treatment, n = 8), and Colitis/SPH (with SPH treatment, n = 8). Mice had ad libitum access to water and food. The experimental colitis model was generated through 1% TNBS cutaneous sensitization and 2.5% TNBS enema challenge phases. TNBS preparation: freshly mixed acetone and olive oil in a 4:1 volume ratio by vortexing rigorously. Then, 4 volume of acetone/olive oil was mixed with 1 volume of 5% (*w*/*v*) TNBS solution in H_2_O to obtain 1% TNBS for sensitization, and 1 volume of 5% TNBS solution was mixed with 1 volume of 70% ethanol to obtain 2.5% TNBS for challenge. The above-prepared solution containing TNBS was replaced with the corresponding solution containing phosphate-buffered saline (PBS) in the Control group. Briefly, on day −8, 150 μL of 1% TNBS solution was applied to the shaved skin area and absorbed rapidly. On day 0, mice were anesthetized with 1.5% isoflurane-mixed gas, given 100 μL of 2.5% TNBS solution by enema using a catheter, and then placed in clean cages for stool collection, weighed and assessed daily. The animals were not fasted prior to the administration of TNBS enema. All above regents were provided by Sigma-Aldrich Corp., St. Louis, MO, USA. On day 4, mice were sacrificed in a CO_2_-euthanization station chamber equipped with an automatic air flow-regulator that is periodically inspected by the Institutional Animal Care and Use Committee (IACUC) at the university and their colon tissues were harvested. Control peptides and SPH (both provided by Hofseth BioCare AS company, Aalesund, Norway) had a similar average molecular weight profile and nutritional calories. The control peptide powder is was dry, water-soluble powder provided by vital protein collagen of bovine source, while SPH is a dry, water-soluble powder which was sourced from salmon. Both powders contained greater than 98% protein and less than 0.5% lipids. Control peptides and SPH were added into drinking water at the final concentration of 2% *w*/*v* and provided to the mice in the Colitis/CP or Colitis/SPH group respectively daily from day −10 to day 4. All experiments were conducted under a protocol approved by Stanford University School of Medicine IACUC in accordance with NIH guidelines (Approval code: A3213-01. Approval date: 19 February 2020 (Through 20 January 2023)).

2.2. Assessment of Disease Activity Index (DAI) [15]: The DAI score, widely used to evaluate the severity of colitis, was determined by summation of the clinical score reported on day 4 after colitis induction. For weight loss, 0: <1%; 1: 1–5%; 2: 6–10%; 3: 11–20%; 4: >20%. For bleeding, 0: negative by using hemoccult (Beckman Coulter, Indianapolis, IN, USA), 2: positive hemoccult; 4: gross bleeding. For stool consistency, 0: normal; 2: slightly loose; 4: liquid.

2.3. Tissue Collection and Preparation: The mice were sacrificed by CO_2_ suffocation, the large intestines were removed together with the cecum and the length of colon was measured in a relaxed position without stretching. The contents of intestines were washed out with cold PBS. The distal parts of colons were fixed in 4% paraformaldehyde, processed, and embedded in paraffin for staining with hematoxylin-eosin (H&E). The lower middle parts of colon tissues were simultaneously frozen and embedded in optimal cutting temperature (OCT) compound media (Tissue-Tek^®^, Sakura Finetek USA, Inc., Torrance, CA, USA), then kept at −80 °C before cryo-sectioning. The upper middle parts were snap frozen and stored at −80 °C for subsequent cytokine and RT-PCR assays. The proximal part of colons was manipulated by hand during the processing steps and not used by this study.

2.4. Histology Study: Specimens were cut into 5 μm sections, deparaffinized, rehydrated and stained with hematoxylin-eosin (H&E) (ScyTek Laboratories, Inc., Logan, UT, USA) according to a standard protocol. Then, H&E-stained colonic tissue sections were scored by a blinded observer according to pathological scoring criteria previously described [16]. Briefly, eight histological components were assessed, including “inflammatory infiltrate”, “goblet cell loss”, “hyperplasia”, “crypt density”, “muscle thickness”, “submucosal infiltration”, “ulcerations”, and “crypt abscesses” (all categorized from 0–3). A total histological severity score from 0 to 24 was obtained by summing the eight item scores.

2.5. Immunohistochemical staining [17]: Specimens frozen in −80 °C embedded in OCT compound media were sectioned (6 µm), and fixed with acetone. A standard immunoperoxidase procedure for immunohistochemistry was used to identify leukocyte (CD4 and CD8 T cells, B cells and macrophages) accumulation using subset-specific monoclonal antibodies (mAbs), then visualized with AEC peroxidase substrate Kit. The sections were counterstained with hematoxylin and mounted in glycergel mounting medium. Then, the numbers of CD4, CD8 and B220 positive stained cells per colonic cross-section area or the percentage of CD68 positive stained areas in total colonic cross-section area were counted in the graph under a 20× light microscope using Image J software (Version 2021, NIH, https://imagej.nih.gov/ij/). All antibodies are listed in Table 1.

2.6. ELISA assays: Colonic tissues were homogenized in Tissue Protein Extraction Reagent (T-PER, Thermo Fisher Scientific, Waltham, MA, USA) with protease inhibitor cocktail (Sigma-Aldrich Corp., St. Louis, MO, USA), and centrifuged 10,000× *g* for 10 min. The levels of cytokine transforming growth factor (TGF)-β1, interleukin (IL)-10, IL-6 and tumor necrosis factor (TNF)-α in the supernatants were measured using ELISA kits (eBioscience, San Diego, CA, USA) following the instructions of the manufacturers.

Keratin-8 (K8) is a major component of the intermediate filaments of single-layered epithelia including enterocytes. The level of protein K8 in the stool has been shown to be significantly increased by intestinal epithelial injury caused by several pathological conditions [18,19]. Therefore, fecal K8 provides a measure of the extent of intestinal epithelial injury. Fecal K8 protein assay was performed according to a modified protocol using anti-K8 antibodies (Troma-1, MilliporeSigma, Temecula, CA, USA). The plates were read at 450 nm using the SpectraMax^®^ i3x multi-mode microplate reader (Molecular Devices, San Jose, CA, USA).

2.7. qRT-PCR assay: The RNeasy mini Kit was used to purify mRNA from mouse colonic tissues that were stored at −80 °C according to the protocol. Then, the sample was added to 40 µL of Buffer GE2 (gDNA elimination buffer) and RNase-free H_2_O to make a final volume of 60 µL. The sample was incubated at 37 °C for 5 min and immediately placed on ice for 2 min. 62 µL of the BC5 Reverse Transcriptase Mix was added to each 60 µL RNA sample for a final volume of 102 µL. The sample was then incubated at 42 °C for exactly 15 min and then the reaction was stopped by heating at 95 °C for 5 min. The cDNA from above was mixed with the RT2 SYBR Green fluor qPCR Mastermix and aliquoted into the wells of the Oxidative Stress RT2 Profiler PCR Array shown below. RT-PCR was performed on an iCycler (Bio-Rad Inc., Richmond, CA, USA). Gene expression was compared using Ct values and the results were calculated using ΔΔ Ct method with normalization to the average expression levels of the housekeeping genes. The anti-oxidative genes and housekeeping gene tested are shown below: FTH1 (ferritin heavy chain 1), HMOX1 (heme oxygenase 1), NQO-1 (NAD(P)H quinone oxidoreductase 1), SOD1 (superoxide dismutase 1, soluble), and GAPDH (glyceraldehyde-3-phosphate dehydrogenase). The above related reagents were purchased from Qiagen, Redwood City, CA, USA.

2.8. Statistical analysis: All data on continuous variables with normal distribution are presented as means ± standard deviation (SD) and analyzed using one-way or two-way analysis of variance (ANOVA) followed by two group comparisons using Newman- Keuls test or Bonferroni post hoc test. Non-normally distributed data are presented as medians and interquartiles (Q1:Q3), and analyzed using non-parametric Kruskal–Wallis test followed by Dunn’s test. *p* < 0.05 was considered statistically significant. All data were analyzed using GraphPad Prism software (Version 5.0, Graphpad Software Inc, San Diego, CA, USA).

## 3. Results

### 3.1. TNBS-Induced Mouse Colitis Could Be Attenuated by SPH

Four groups of mice were treated as indicated in Figure 1. As shown in Figure 2A, the TNBS treatment caused significant loss of body weight from day 1 through day 4 in the Colitis, Colitis/CP and Colitis/SPH groups (*p* < 0.01) when compared to the Control group. By the end of the study (on day 4), mice in three groups had significant weight loss, increased DAI scores, and shortened colon length when compared to the Control group as shown in Figure 2B,D. However, these changes were alleviated in the Colitis/SPH group, suggesting that SPH treatment can attenuate weight loss, colon shortening, and decrease DAI in TNBS-induced mouse colitis model compared with control peptide.

### 3.2. SPH Ameliorates Colonic Tissue Injury in TNBS-Induced Colitis Model

The H&E-stained histological sections (Figure 3A) showed intact colonic mucosa, crypts, stroma, and submucosa, limited inflammatory cell infiltration in the submucosa, and no goblet cell loss, muscle thickening, or crypt abscess and ulceration in the Control group. After TNBS treatment, the mice in the Colitis and Colitis/CP groups showed intense inflammatory lesions, including severe epithelial cell injury, crypt hyperplasia, increased macroscopic spaces between crypts, submucosal edema, obvious muscle thickening, goblet cell loss, and inflammatory cellular infiltration in the submucosa. However, SPH treatment ameliorated colonic tissue injury induced by TNBS with a relative intact surface epithelium, mild submucosal edema and crypt hyperplasia, mild muscle thickening and reduced goblet cell loss and inflammatory cell infiltration. For the histology score (Figure 3B), the Colitis and Colitis/CP groups showed much higher scores than the Control group and Colitis/SPH (*p* < 0.01), and SPH treatment decreased the histology score compared to the Colitis/CP group (*p* < 0.01).

In addition to the tissue damage observed on histopathology, we also measured significant changes in the intestinal epithelial injury biomarker K8 in stool samples. While the level of K8 was increased in the Colitis/SPH group (*p* < 0.05) compared to the Control group, the increase was significantly attenuated compared to the Colitis and Colitis/CP groups (*p* < 0.01) (Figure 3C). Taken together, these results suggests that SPH treatment significantly diminishes colonic epithelial injury as determined by quantitative K8 shed in stool samples.

### 3.3. SPH Confines Gut Inflammation in Mice with TNBS-Induced Colitis

To elucidate the protective mechanisms of SPH in improving colitis, especially the anti-inflammatory effect, we determined the expressions of relevant immune cell membrane markers and the levels of several cytokines in colon tissue by IHC and ELISA, respectively. As demonstrated in Figure 4, the infiltration of CD4^+^ T, CD8^+^ T and B220^+^ B lymphocytes in colon tissue were significantly increased in Colitis and Colitis/CP groups compared with the Control group (*p* < 0.01). SPH treatment significantly reduced the infiltration of lymphocytes in the colon tissues (*p* < 0.01). Intriguingly, there was no statistically significant difference in the CD68^+^ expression of macrophages among four groups as shown in Figure 5 (*p* > 0.05). In addition, as depicted in Figure 6, the levels of pro-inflammatory cytokines TNF-α and IL-6 increased (*p* < 0.01) and the levels of anti-inflammatory cytokines IL-10 and TGF-β1 showed no or minimal increase in the Colitis and Colitis/CP groups compared with the Control group. Intriguingly, SPH treatment prevented a significant increase in pro-inflammatory TNF-α and IL-6 and additionally augmented the production of anti-inflammatory IL-10 and TGF-β1 compared with the other two colitis groups (*p* < 0.01). Combined with the infiltration of lymphocytes and the patterns of pro- and anti-inflammatory cytokines in colon tissue, together these results suggest that SPH inhibits gut inflammation and may play an immunoregulatory role in the TNBS-induced colitis model.

### 3.4. SPH Upregulates the Expressions of Anti-Oxidative Genes in TNBS-Induced Colitis

To further investigate the mechanism by which SPH may be exerting its protective effects, the expression of anti-oxidative stress relevant genes such as FTH1, HMOX1, NQO-1, and SOD1 were determined. TNBS treatment marginally upregulated the expression of HMOX1, but not the other oxidative stress-related marker genes. SPH treatment led to the upregulation of oxidation protective genes including FTH1, HMOX1, NQO-1, and SOD compared with the experimental groups (shown in Figure 7), suggesting that the protective role of SPH is associated with anti-oxidative gene induction.

## 4. Discussion

The present study demonstrates that SPH provides robust and effective protection in a pre-clinical TNBS-induced colitis model of IBD. Together, the data demonstrate a significant reduction in macroscopic injury (colon length and DAI) and clinical severity (weight loss). Consistent with prior work by our group member Framroze, there was also evidence of the canonical heme oxygenase (HO) anti-oxidant pathway induction by the SPH in vivo [20]. Moreover, the reduction in injury and induction of HO was accompanied by a significant reduction in inflammatory cell infiltration, an important hallmark of IBD associated immune response.

In recent years, the immunological dysregulation of and associated immunotherapies for IBD have received extensive attention [21]. The immune system is divided into innate immunity and adaptive immunity, both of which are involved in the pathogenesis of IBD [3]. In IBD, tissue damage occurs in areas heavily infiltrated with activated CD4^+^ T lymphocytes, and Th1 cell-derived cytokines are important mediators of tissue damage in Crohn’s disease, while Th2-type cytokine IL-13 may play an essential role in ulcerative colitis [22]. However, in both Crohn’s disease and ulcerative colitis, there is a marked increase in the local synthesis of various downstream nonspecific inflammatory mediators, such as free radicals, leukotrienes, chemokines, and pro-inflammatory cytokines, which, along with inflammatory cells, interact with and amplify signals that ultimately culminate in tissue damage [3,22]. It has been demonstrated that intrarectal administration of TNBS to mice can induce transmural colitis, primarily driven by a Th1-mediated immune response and characterized by lamina propria infiltration of T cells, B cells, macrophages, dendritic cells and neutrophils. TNBS colitis has therefore been extensively used in the study of immunologic aspects relevant to IBD, including cytokine secretion patterns and the effects of potential immunotherapies [23].

To elucidate the mechanisms of immunologic modulation by SPH in TNBS-induced colitis, the infiltration of relevant immune cells and the level of associated cytokines in TNBS-damaged colon tissues were investigated. The results showed that CD4^+^ T, CD8^+^ T and B220^+^ B lymphocytes were significantly increased in the Colitis and the Colitis/CP group, suggesting that these lymphocytes are involved in the pathogenesis of the experimental colitis. In contrast, SPH treatment reduced the infiltration of CD4^+^ T, CD8^+^ T and B220^+^ B lymphocytes and the levels of pro-inflammatory cytokines TNF-α and IL-6 in colon tissues compared with control peptide. This suggests that the reduction of gut inflammation by SPH in the TNBS colitis model includes an immunomodulatory effect.

Intestinal resident macrophages expressing the common myeloid marker CD68 are the most abundant mononuclear phagocytes in the intestine. These cells play a critical role in maintaining intestinal homeostasis and are also drivers of the pathology associated with IBD [3,24,25]. Patients with Crohn’s disease display defective innate immune responses and present with an inflammatory macrophage population that produces large amounts of pro-inflammatory cytokines such as TNF-α and IL-6 [1]. Several studies have found that the levels of TNF and IL-6 are increased in the serum and the intestinal mucosa of patients with active Crohn’s disease, and are positively correlated with the clinical disease activity and histopathological signs of inflammation in Crohn’s disease patients [3,26,27,28,29,30]. TNF-α and IL-6 also play significant roles in ulcerative colitis pathogenesis, because an increase in TNF-α expression may contribute to defective mucosal barrier function in ulcerative colitis patients and exacerbate inflammation, while an increase in IL-6 plays a significant role in the trans-signaling process and chronic inflammation of ulcerative colitis [31]. Despite the significant difference in measured cytokines between treatment and control groups in the present study, there was no statistically significant difference in the number of CD68^+^ macrophages among the four groups in the present study. It is plausible that additional studies could determine whether there is a difference in the function or activity of tissue macrophages between the groups.

In addition to pro-inflammatory cytokines, anti-inflammatory cytokines play an important role in maintaining the immune homeostasis. IL-10, which is secreted by regulatory macrophages and T cells, inhibits antigen-presenting cell activation and Th1 and Th2 cell responses, while enhancing the differentiation of regulatory T cells (Tregs). IL-10 is a key anti-inflammatory cytokine in the maintenance of mucosal immunological tolerance, and both mice and humans develop spontaneous IBD when IL-10 or its receptors are genetically disrupted [32]. In addition, the therapeutic efficacy of anti-TNF agents to manage intestinal inflammation of IBD is critically dependent on IL-10 signaling [32]. TGF-β is an immune-suppressive cytokine produced by many cell types including epithelial cells, immune cells, and fibroblasts. Active TGF-β binds to its receptor and regulates mucosal immune reactions through the TGF-β signaling pathway [26]. TGF-β plays important roles in maintaining intestinal homeostasis and regulating T cell differentiation, and protecting against the development of spontaneous colitis [26]. Moreover, Tregs suppress colonic inflammation by down-regulating Th1 and Th17 responses in the presence of IL-10 and TGF-β [3]. The results presented in this study showed that the level of anti-inflammatory cytokines IL-10 and TGF-β1 decreased in the Colitis and Colitis/CP groups, while SPH treatment elevated the levels of IL-10 and TGF-β1 compared with the control peptide. This result seems to indicate that SPH may play an immunoregulatory role in TNBS-induced colitis. Although the number of macrophages did not change significantly, a finding inconsistent with other studies [25], their function may have been influenced by IL-10 and TGF-β1 [3], and could have polarized to M_2_ anti-inflammatory macrophages following SPH treatment.

Oxidative stress (OS), a phenomenon caused by an imbalance between the production and accumulation of reactive oxygen species (ROS) in cells and tissues, has been recognized as a contributing pathologic feature for many diseases, including IBD [4]. High levels of ROS damage important cellular structures like proteins, lipids, and nucleic acids. Persistent redox imbalance can lead to irreversible damage to cellular function, and eventually cell death and disease [33,34]. The gastrointestinal tract is susceptible to ROS induced injury as a result of the fermentation of various dietary substances, resident immune cell activation, and intestinal flora-induced inflammation, all of which serve as potential sources of ROS [35]. It is well known that the excess production of ROS in intestinal mucosal cells can induce immune responses that damage intestinal epithelial cells, disrupt the integrity of the intestinal barrier, and trigger intestinal inflammation. Previous research has demonstrated that the level of malondialdehyde (MDA), a lipid peroxidation product, is significantly increased in colon tissues after the administration of TNBS [4,36]. Although intestinal epithelial cells have developed numerous antioxidant defense measures, when excessive damage or ineffective repair occurs, subsequent cell injury and death is triggered. Antioxidants have therefore been suggested as a viable therapeutic approach for attenuating tissue damage induced by oxidative stress [35].

Dietary antioxidants and consumption of antioxidant-rich foods has specifically been shown to modulate OS-related gene expression particularly within the gastrointestinal tract [34]. In this study, we found that SPH treatment did indeed upregulate the expression of oxidative protective genes, including FTH1, HMOX1, NQO-1, and SOD, compared with control peptide. The FTH1 gene encodes the heavy subunit of Ferritin (H-Ferritin) which sequesters dietary free iron in a soluble and nontoxic state and hence is an important member of the antioxidant system. Previous studies have shown that oxidative stress increases the synthesis of ferritin subunits which reduces the accumulation of ROS [37], and that the upregulation of ferritin heavy chain by NF-κB (nuclear factor kappa light chain enhancer of activated B cells) can inhibit TNFα-induced apoptosis by suppressing ROS [38]. The HMOX1 gene encodes the enzyme HO-1 which catalyzes the rate-limiting step in heme degradation and generates equimolar amounts of iron ions, biliverdin and carbon monoxide. HO-1 and its products have cytoprotective properties and regulate important biological processes including oxidative stress, inflammation, apoptosis, cell proliferation, fibrosis, and angiogenesis [39]. As such, the ability to upregulate HO-1 expression is an important protective factor for gastrointestinal health and a potential therapeutic target in gastrointestinal diseases [40]. Heme iron, but not inorganic iron, promotes ROS production, and oxidative DNA damage. This results in a preferential cytotoxicity in non-malignant intestinal epithelial cells, and HO-1 confers protection against the detrimental effects of hemin [41]. Ferritin synthesis is correlated with HO-1 expression in cultured cells and heme-derived iron may play an essential role in the induction of ferritin synthesis [42]. However, the cytoprotective effect of HO-1 requires the co-expression of FTH, which controls the pro-oxidant effect of labile iron released from the protoporphyrin IX ring of heme, and HO-1 exerts only partial protection against oxidative stress in cells lacking H-Ferritin, indicating the cooperation of both enzymes [43,44]. The NQO1 gene encodes NAD(P)H: Quinone Oxidoreductase 1, which is widely distributed in organs, especially in the liver, kidney, and gastrointestinal tract, can be induced by various chemicals. NQO1 enzyme plays an important role in the detoxification metabolism of the body and protection against oxidative damage [45]. SOD is an enzymatic antioxidant that catalyzes the conversion of O_2_^·−^ to H_2_O_2_ and helps maintain the redox balance by diffusing the superoxide in oxidative stress-induced pathology [35].

It has previously been shown that the protein hydrolysate from plant or fish could serve as powerful antioxidants in the treatment of diseases associated with oxidative stress through decreasing the levels of ROS and MDA, increasing intracellular antioxidant enzymes of SOD, glutathione peroxidase (GPx), and catalase (CAT), and triggering nuclear translocation of nuclear factor erythroid 2-related factor (Nrf2) and the expression of HO-1 [46,47]. The findings herein extend those prior observations on the effects of SPH in lowering oxidative stress and immune modulation in a pre-clinical model of IBD.

This is the first study about the protective role of SPH in IBD mouse model. However, the primary mechanism of the protective effect of SPH is unknown. As we know, proteins are digested into free amino acids, dipeptides, and tripeptides in the digestive tract, and are absorbed in this form mainly in the small intestine. Food is an essential factor in the regulation of the renewal of gastrointestinal mucosa. Food intake can stimulate cell proliferation and the growth and renewal of gastrointestinal mucosa. In the case of fasting or parenteral nutrition, the renewal of gastrointestinal mucosa is inhibited leading to its atrophy or hypoplasia [48,49], while refeeding leads to a rapid increase of cell proliferation in the gastrointestinal tract and the restoration of its structure [50,51] due to the reflex of the enteric nervous system and the release of gastrointestinal hormones that stimulate the growth of the colon, in which gastrin produced and released by G cell in the stomach plays a special role [52]. The presence of small peptides and aromatic amino acids in the stomach is a main factor that acts directly on the G cells and stimulates the release of gastrin [52,53]. Although digestion of proteins in the digestive tract begins within the stomach with participation of pepsin, the essential digestion of proteins take place in the small intestine with the participation of pancreatic and intestinal peptidases after leaving the stomach. Moreover, pepsin is an endopeptidase that digests internal peptide bonds [54]. Therefore, the release of amino acids and short peptides in the stomach from food is limited. SPH is a product of enzymatic protein hydrolysis and contains a high concentration of amino acids and short peptides which may lead to a significant release of gastrin stimulating the renewal of colonic mucosa, increasing the mucus production in the intestine, and improving the barrier function of the intestinal mucosa. This concept is consistent with previous observations that also ghrelin, another factor that stimulates the growth of gastrointestinal mucosa, exhibits a protective and therapeutic effect in experimental colitis evoked by TNBS [55,56], dextran sulfate sodium (DSS) [57,58], and acetic acid [59,60].

Above is our hypothesis regarding the primary mechanism of SPH, which should be verified in future research. There are also other limitations to this study, such as the difference in the phenotype and function of tissue macrophages between the groups, and the protein expression levels and activity of antioxidant HO-1, SOD, NQO-1 and FTH1. In the future, we can conduct in-depth research from these aspects. In addition, although the animal models are useful in the research concerning etiopathogenesis of IBD and new therapeutic concepts in this disease, there are numerous differences between animal models and clinical IBD. Therefore, the results obtained with animal models may or may not be consistent with the clinical effect, and need to be verified in clinical study.

## 5. Conclusions

In conclusion, SPH treatment significantly reduces colon tissue injury in the mouse TNBS acute colitis model by alleviating inflammatory cell infiltrates and oxidative stress injury. This suggests that the peptides in SPH have a potential immunomodulatory action through the upregulation of protective, antioxidant gene pathways including HMOX1. These effects limited macroscopic colonic damage and supported general health assessed by weight loss. In contrast, control peptides had no beneficial effect, ruling out enhanced nutrition as a means of protecting against the effects of acute colitis. It may be concluded from our results that SPH peptides exert an anti-inflammatory and anti-oxidative effect in the mouse TNBS-colitis model of IBD.

## Figures and Tables

**Figure 1 biomolecules-12-01287-f001:**
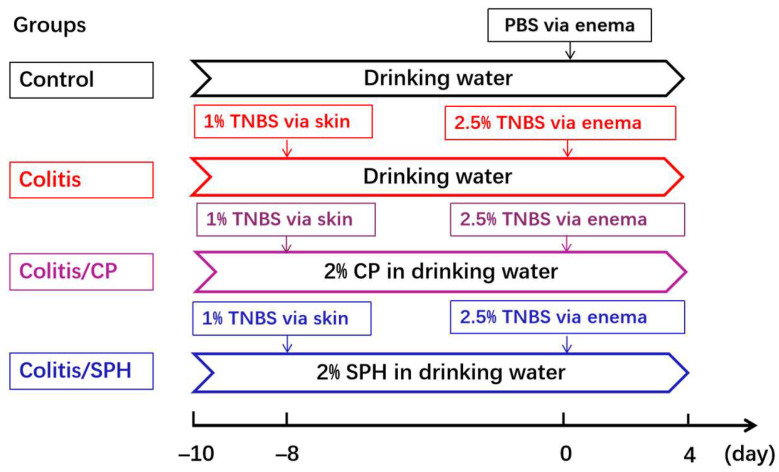
Experimental design for acute TNBS-induced mouse colitis model. Colitis (n = 8) was induced by cutaneous sensitization with 1% TNBS followed by enema challenge with 2.5% TNBS. Control peptides and SPH were provided to the mice in the Colitis/CP (n = 8) or Colitis/SPH (n = 8) group, respectively, by drinking water daily at a final concentration of 2% *w*/*v* from day −10 to day 4. The intervention solution in the Control group (n = 7) had PBS instead of TNBS. Note: TNBS: 2,4,6-trinitrobenzene sulfonic acid. CP: control peptide. SPH: Soluble protein hydrolysate. PBS: phosphate puffered saline. Note: we actually set up two other groups (Colitis/CP and Colitis/SPH) in the pilot experiments, but it was found that SPH and CP had no obvious negative effects on the intestinal tissues or body weight of these mice, so they were omitted in the current study.

**Figure 2 biomolecules-12-01287-f002:**
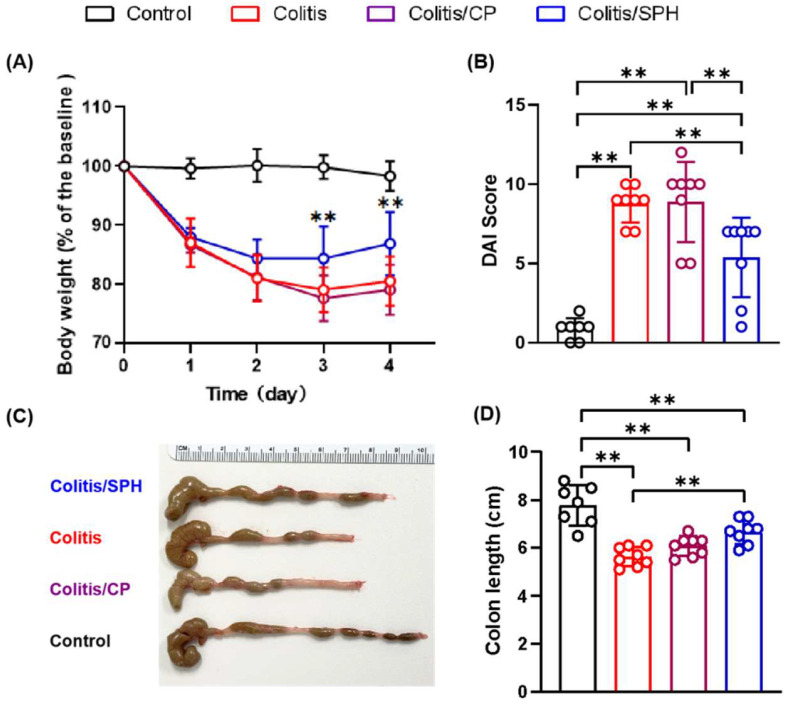
Protective role of SPH in TNBS-induced colitis model. (**A**) Longitudinal body weight change of the baseline (%). Note that except for the Control group (n = 7), the body weights in the other three groups (all n = 8) on days 1–4 were significantly lower than those on day 0 (*p* < 0.01), and the changes in body weight of the baseline were higher than that in the Control group at each time point (*p* < 0.01). **: SPH treatment attenuated weight loss on day 3 and 4 compared to Colitis and Colitis/CP groups (*p* < 0.01). (**B**) Disease activity index (DAI) scores. (**C**) Representative macroscopic images of colon changes. (**D**) Comparison of colon length. Data are presented as mean ± SD and analyzed by one-way ANOVA (**B**,**D**) or two-way ANOVA test (**A**). **: *p* < 0.01.

**Figure 3 biomolecules-12-01287-f003:**
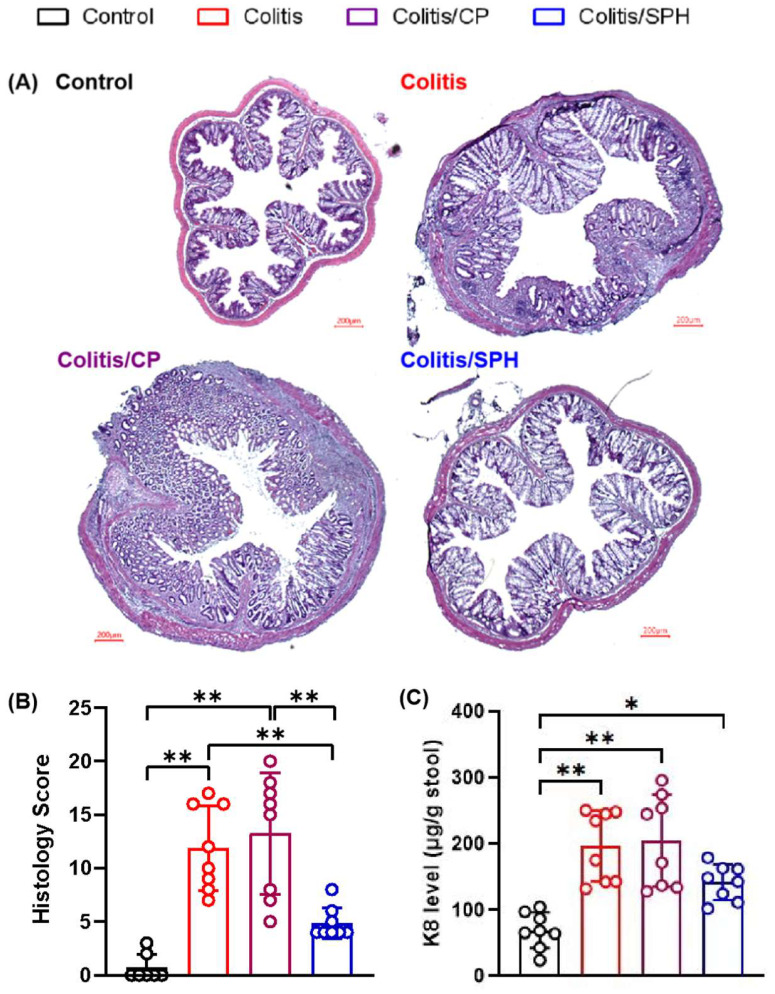
SPH reduces tissue injury of the colon in the TNBS-induced colitis model. (**A**) Histology images with H&E stain (×200). (**B**) Histology grading on gastrointestinal injury. (**C**) Keratin (K8) level in stool by ELISA. The numbers of mice in the four groups were 7, 8, 8, and 8, respectively. Data are presented as mean ± SD and analyzed by one-way ANOVA test. **: *p* < 0.01, *: *p* < 0.05.

**Figure 4 biomolecules-12-01287-f004:**
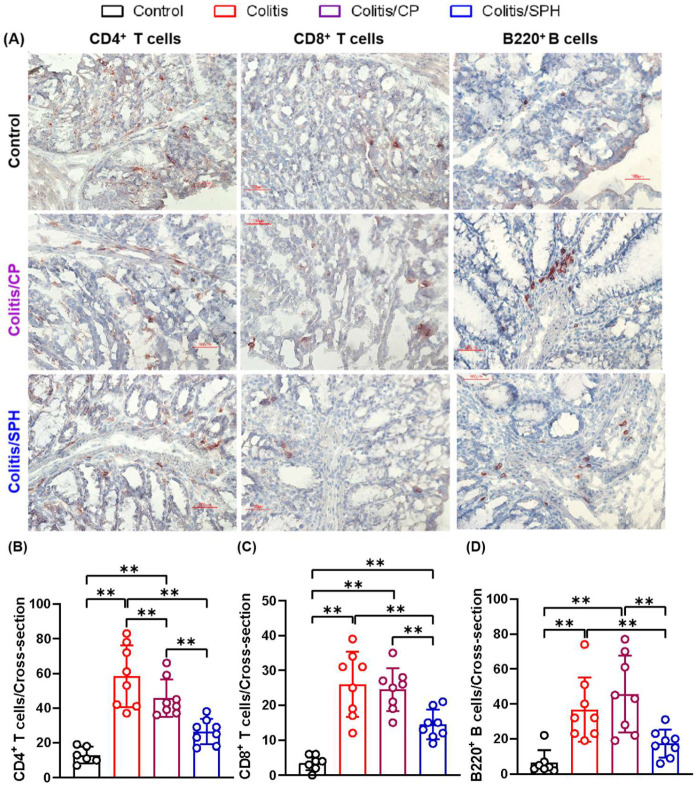
SPH reduces the infiltration of lymphocytes in TNBS-induced colitis model. (**A**) IHC images (×200) and (**B**–**D**) the numbers of CD4-, CD8- and B220-positive stained cells per colonic cross-section area. The infiltration of CD4^+^ T, CD8^+^ T and B220^+^ B lymphocytes in colon tissue were significantly increased in the Colitis (n = 8) and Colitis/CP (n = 8) groups compared with the Control group (n = 7), while SPH treatment (n = 8) significantly reduced the infiltration of lymphocytes in the colon tissues. Data are presented as mean ± SD and analyzed by one-way ANOVA test. **: *p* < 0.01.

**Figure 5 biomolecules-12-01287-f005:**
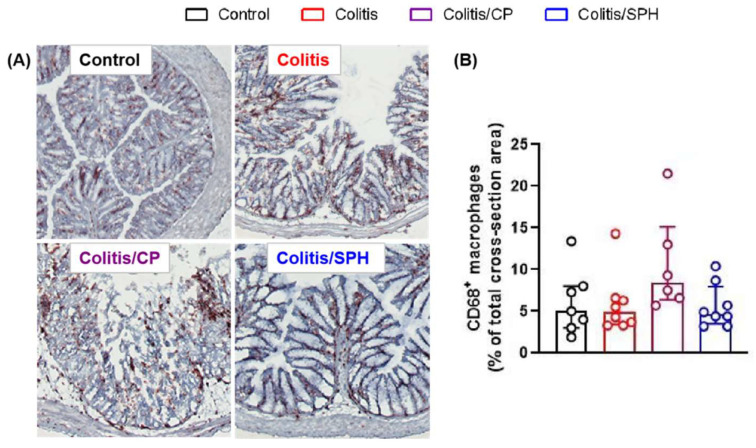
The infiltration of macrophages in colon tissues under different interventions. (**A**) IHC images (×200) and (**B**) percentage of CD68-positive stained areas in total colonic cross-section area. The numbers of mice were 7, 8, 8, and 8 in each individual groups. Data are presented as median and interquartiles (Q1:Q3) and analyzed by Kruskal–Wallis test. Note that there was no statistically significant difference in the CD68^+^ expression of macrophages among four groups (*p* > 0.05).

**Figure 6 biomolecules-12-01287-f006:**
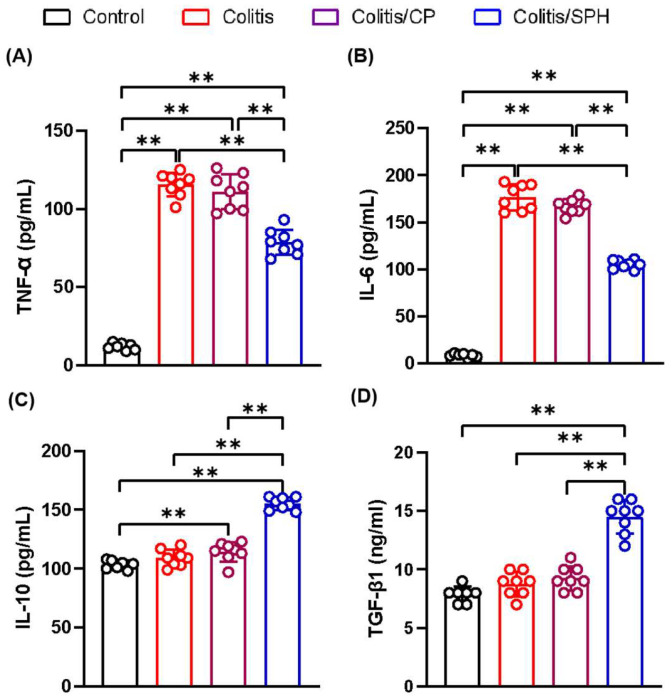
SPH changes the secretion patterns of pro- and anti-inflammatory cytokines in the TNBS-induced colitis model. (**A**–**D**) The levels of tumor necrosis factor (TNF)-α (**A**), interleukin (IL)-6 (**B**), IL-10 (**C**) and transforming growth factor (TGF)-β1 (**D**) in colon tissues homogenate detected by ELISA. Compared with the Control group (n = 7), the levels of pro-inflammatory cytokines TNF-α and IL-6 increased and the levels of anti-inflammatory cytokines IL-10 and TGF-β1 decreased in the Colitis (n = 8) and Colitis/CP (n = 8) groups, while SPH treatment (n = 8) reduced the levels of TNF-α and IL-6 and elevated the levels of IL-10 and TGF-β1. Data are presented as mean ± SD and analyzed by one-way ANOVA test. **: *p* < 0.01.

**Figure 7 biomolecules-12-01287-f007:**
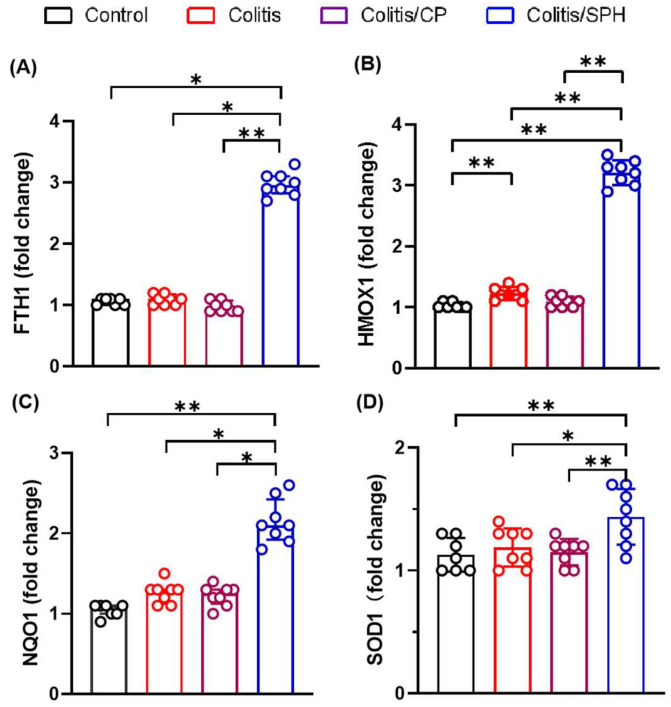
SPH upregulates the expressions of anti-oxidative genes in TNBS-induced colitis model. (**A**–**D**) Assessment of the mRNA expressions of FTH1 (ferritin heavy chain 1), HMOX1 (heme oxygenase 1), NQO-1 (NAD(P)H quinone oxidoreductase 1), and SOD1 (superoxide dismutase 1, soluble) genes in colon tissues using RT-PCR. The numbers of mice in the four groups were 7, 8, 8, and 8, respectively. Data are presented as mean ± SD and analyzed by one-way ANOVA test in (**B**,**D**), while median and interquartiles (Q1:Q3) and Kruskal–Wallis test were used in (**A**,**C**). **: *p* < 0.01, *: *p* < 0.05.

**Table 1 biomolecules-12-01287-t001:** Sources and working conditions for immunohistochemical reagents [16].

Reagent	Manufacture	Catalog #	Clone #	Working Solution (μg/mL)	Incubation Time at Room Temperature (Minutes)
Rat anti-mouse CD4 mAb	Biolegend, San Diego, CA, USA	100402	GK1.5	2.5	60
Rat anti-mouse CD8 mAb		100702	53–6.7	2.5	60
Rat anti-mouse B220 mAb		103201	RA3–6B2	2.5	60
Rat anti-mouse CD68 mAb		137002	FA-11	2.5	60
Biotin-goat anti-rat lgG Ab	Jackson ImmunoResearch Laboratories, Inc., West Grove, PA, USA	112-065-062	N/A	4.0	30
HRP-streptavidin conjugate		016-030-084	N/A	5.0	30
AEC peroxidase substrate Kit	Vector Laboratories, Inc., Newark, CA, USA	SK-4200	N/A	N/A	10–20

mAb: monoclonal antibody.

## Data Availability

Not applicable.

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
