# Peer review of "Soluble Protein Hydrolysate Ameliorates Gastrointestinal Inflammation and Injury in 2,4,6-Trinitrobenzene Sulfonic Acid-Induced Colitis in Mice"

_biomolecules, 2022, doi:10.3390/biom12091287_

Round 1

Reviewer 1 Report

This is a very well-planned and well-executed study with valuable results, in which it was found that the SPH protein solution has a protective effect against the colitis-inducing effect of TNBS in an animal model. However, taking into account some aspects, the content of the article needs improvement.

1. In section 2.1, the number of animals used in the experimental groups must be indicated.

2. In part B of figure 1, it must be indicated which diagram indicates which group.

3. In section 3.1, the body weight of the animals must be given in grams.

4. In section 3.4, I consider it necessary not only to measure the mRNA expression of antioxidant genes but also to measure the functional activity of at least one antioxidant gene (e.g., SOD). By itself, the change in gene expression does not mean a functional consequence.

5. How was the purity of the SPH solution used determined? How to prove that there was no DNA or LPS contamination in the SPH solution? It is known that DNA sequences or LPS can reduce the activity of chemical colitis via TLRs. It would be worthwhile to address this aspect in the discussion as well.

After the above-mentioned corrections, I recommend accepting the article for publication.

Author Response

Responses to Reviewer 1 comments:

Dear reviewer, thank you very much for your timely and valuable comments, please find the point-by-point responses as follows:

Point 1: In section 2.1, the number of animals used in the experimental groups must be indicated.

Response 1: According to the experimental procedure as shown in Figure 1, 31 mice were randomly divided into four groups: Control (7), Colitis (8), Colitis/Control peptide (CP) (8), and Colitis/SPH (8).

Point 2: In part B of figure 1, it must be indicated which diagram indicates which group.

Response 2: We used different color to indicate different group, such as black indicate control group, red, purple and blue respectively indicates colitis, colitis/CP and colitis/SPH group. The color codes are described on the top of the figure.

Point 3: In section 3.1, the body weight of the animals must be given in grams.

Response 3: We totally agree with the reviewer that it would be much better to use absolute values of body weight in grams to generate the figure. But, for your information, figure 2A was made from two sets of separated experiment A (n=15) and B (n=16), about two months apart. The average body weight is 27.5g in A-exp and 24.2g in B-exp. For better comparison in each experiment, we carefully choose them with very similar body weight by removing some smaller or bigger ones as the mice are frequently varied in body weight even they are at same age or from same litter.  The data of absolute body weight (in grams) either from A- or B-experiment is pretty good, however it’s not so good when we mixed the two sets of results together due to the difference of average body weight between A and B experiments. Therefore, we have to normalize each body weight with percentage change over their initial value at day-0 in order to properly combine the two sets of results for better comparison.

Point 4: In section 3.4, I consider it necessary not only to measure the mRNA expression of antioxidant genes but also to measure the functional activity of at least one antioxidant gene (e.g., SOD). By itself, the change in gene expression does not mean a functional consequence.

Response 4: We agree with the reviewer that it would be perfect if we can correlate the up-regulated mRNA levels with final functional activities of their end-product proteins (enzymes). This is an extremely profound and insightful comment, but the present study was focused on the efficacy of SPH pretreatment in TNBS-induced colitis through different approaches such as biochemical, immunological, pathological analyses. Indeed, the overexpressed mRNA levels of antioxidant genes are very encouraging, which led us toward extensive molecular mechanism studies through investigating cellular antioxidative signaling pathways, especially for Keap1-Nrf2-ARE signaling. These studies are still ongoing and definitely the recommended functional activity assays for the antioxidant gene’s end-products will be included, but all the mechanism-focused new results will be separately reported in near future and hopefully by the same journal. On the other hand, a previous placebo-controlled study (led by our coauthor Bomi Framroze) has reported that the FTH1 gene-associated end-product ferritin was unregulated after SPH-treatment as evidenced by a commensurate increase in its serum protein level (see ref below).

Framroze Bomi, Sanjay Vekariya and Swaroop Dhruv. A Placebo-Controlled Study of the Impact of Dietary Salmon Protein Hydrolysate Supplementation in Increasing Ferritin and Hemoglobin Levels in Iron-Deficient Anemic Subjects. J. Nutr.Food.Sci 2015, 5:4. (DOI: 10.4172/2155-9600.1000379)

Point 5: How was the purity of the SPH solution used determined? How to prove that there was no DNA or LPS contamination in the SPH solution? It is known that DNA sequences or LPS can reduce the activity of chemical colitis via TLRs. It would be worthwhile to address this aspect in the discussion as well.

Response 5: Each batch SPH is measured for protein purity (98%+) by Eurofins, since SPH is registered and sold under a oral use NDI in US FDA and Health Canada.
Both DNA and LPS (lipopolysaccharides) show as below the LOD established for oral use.

Reviewer 2 Report

Manuscript # biomolecules-1871769

Title: Soluble Protein Hydrolysate ameliorates gastrointestinal inflammation and injury in TNBS-induced mouse colitis model

Authors: Wei et al.

The above manuscript is interesting; the authors tested the oral administration of Soluble Protein Hydrolysate, produced by the enzymatic hydrolysis of salmon offcuts on the development of 2, 4, 6-trinitrobenzene sulfonic acid (TNBS)-induced colitis in mice. However, it should be noted that the manuscript contains some deficiencies and errors that must be removed before accepting the manuscript for publication.

List of the errors and deficiencies:

  1. Abstract, line 13, Introduction, line 58. The statement that “There is no cure for IBD” is not true. As the authors reported in the next lines, numerous methods of treating IBD are used. The problem is the lack of methods for a complete and permanent cure of this disease. Therefore “There is no cure for IBD” should be replaced with “There is no effective therapy for a complete and permanent cure for IBD”.
  2. Abstract, line 22 and section “Materials and Methods”. The authors should at lest 5 experimental groups: (1) Control; (2) Soluble Protein Hydrolysate (SPH) without colitis; (3) Colitis; (4) Colitis +Control peptide (CP); (5) Colitis + SPH. A group SPH without colitis is necessary in order to know whether SPH is causing negative effects in the colon and other organs of the body.
  3. Abstract, line 24. The authors should provide the form of SPH administration and time relation between TNBS and SPH administration.
  4. Abstract, lines 28-29. What does “SPH pre-treatment improved the DAI score and tissue injury” mean? Authors state should unequivocally whether administration of SPH increased or decreased DAI and tissue damage. This comment also applies to the Results and Discussion sections.
  5. Abstract, line 31 and next parts of the manuscript. All abbreviations should be presented in their full name at the point where they appear for the first time. Full names of abbreviation presented in the Abstract should be repeated in the body of the manuscript at the place of the first use, as well as in Tables and Figure legends. Tables and Figures should be understandable without carefully reading the text of the manuscript. On the other hand, the number of abbreviations should be reduced. The excess of abbreviations makes the manuscript difficult to understand.
  6. Materials and Methods, line 82 and next pats of the manuscript. In the case of all drugs, regents, animals, assays and equipment, the authors should provide general and trade name of the reagent or equipment, manufacturer’s name, city, and country.
  7. Materials and Methods, line 87. Since 1975 is not available on the market.
  8. Materials and Methods, line 85. The authors stated That “Mice had ad libitum access to water and food”. Were the animals fasted prior to the administration of TNBS enema?
  9. Material and Methods. How did the authors prepare 2.5% TNBS solution? The same vehicle should be used in control and SPH without colitis groups. All these data should be presented in the manuscript.
  10. Material and Methods, line 90. The authors should in detail present haw the animals were euthanized.
  11. Material and Methods, line 92. The authors should present more details on control peptides such as their biological source, characteristics, average molecular weight, and form provided by the manufacturer. The same information should be provided for SPH. In addition, in the case of SPH, the authors should declare whether SPH contained only peptides and proteins, or it also contained lipid admixture.
  12. Material and Methods, lines 95-97. Authors should provide the protocol approval number, as well as the date of its issue and the expiry date of its validity.
  13. Figure 1. Figure 1A concerns study design and for this reason should be presented in Materials and Methods. However, Figure 1B-D presents result of the study and for this reason should be presented as a separate Figure in the section Results.
  14. Figure legends. The authors should provide the number of observations in each experimental group and state how the results are expressed in the bars.
  15. Figure 2-6. Figure legends are not for presenting the interpretation of results, but they are for presenting what is shown in the figures.
  16. Discussion. The authors should present the importance of animal model of IBD and their limitations (PMID: 27206158; PMID: 34938152). These models are useful in the research concerning etiopathogenesis of IBD and new therapeutic concepts in this disease. It should be noted, however, that there are numerous differences between these models and clinical IBD. Therefore, the results obtained with these models may or may not be consistent with clinical effect.
  17. Discussion. The authors presented some mechanism involved in the protective/therapeutic effect of SPH in TNBS-induced colitis. However, the authors should state that the primary mechanism of protective effect of SPH is unknown. Proteins in the digestive tract are digest into free amino acids, dipeptides, and tripeptides, and in this form are absorbed, which takes place mainly in the small intestine. Thus, there is not direct effect of SPH on the mucosa in the colon. The authors should present the concept that the protective effect of SHP is probably due to the presence of amino acids and short-chain peptides in SPH. Food is essential factor in the regulation of the renewal of gastrointestinal mucosa. Food intake stimulates cell proliferation and the growth and renewal of gastrointestinal mucosa. In the case of fasting, the renewal of gastrointestinal mucosa is inhibited leading to its atrophy (PMID: 8527816; PMID: 3710060), and the degree of weight loss in the digestive tract organs is greater than the body weight loss (PMID: 5659346). Also, in the case of parenteral nutrition, when there is no food in the gastrointestinal tract, the cell renewal of the gastrointestinal mucosa is inhibited, and hypoplasia develops (PMID: 4214726). On the other hand, refeeding leads to a rapid increase in cell proliferation in the gastrointestinal tract and the restoration of its structure (PMID: 1664265; PMID: 6142558). The mechanisms stimulating the renewal of the mucosa after food intake are mainly based the reflex of the enteric nervous system and release of gastrointestinal hormones. Among the gastrointestinal hormones that stimulate the growth of the colon, gastrin plays a special role (PMID: 25716961). Gastrin is produced and released by G cell in the stomach. The presence of small peptides and aromatic amino acids in the stomach is a main factor that acts directly on the G cells and stimulates the release of gastrin (PMID: 6806140; PMID: 25716961). Digestion of proteins in the digestive tract begins within the stomach with participation of pepsin. However, it should be stated that the essential digestion of proteins take place in the small intestine, after leaving the stomach, with the participation of pancreatic and intestinal peptidases. Moreover, pepsin is an endopeptidase that digests internal peptide bonds (PMID: 20522896). Therefore, the release of amino acids and short peptides in the stomach from food is limited. On the other hand, SPH is a product of enzymatic protein hydrolysis and most likely contains a high concentration of amino acids and short peptides, which leads to a significant release of gastrin, thereby significant stimulating the renewal of colonic mucosa and increasing the mucus production in this intestine. This concept is consistent with previous observations that also ghrelin, another factor that stimulates the growth of gastrointestinal mucosa, exhibits a protective and therapeutic effect in experimental colitis evoked by TNBS (PMID: 16697735; PMID: 19617644), dextran sulfate sodium (DSS) (PMID: 23549326; PMID: 25634696), as well as acetic acid (PMID: 26769837; PMID: 28538694). After presenting this concept with appropriate references, the authors should state this hypothesis should be verified in future research.

Author Response

Responses to Reviewer 2 comments:

Dear reviewer, thank you very much for your timely and valuable comments, please find the point-by-point responses as follows: 

Point 1: Abstract, line 13, Introduction, line 58. The statement that “There is no cure for IBD” is not true. As the authors reported in the next lines, numerous methods of treating IBD are used. The problem is the lack of methods for a complete and permanent cure of this disease. Therefore “There is no cure for IBD” should be replaced with “There is no effective therapy for a complete and permanent cure for IBD”.

Response 1: We have revised it.

Point 2: Abstract, line 22 and section “Materials and Methods”. The authors should at lest 5 experimental groups: (1) Control; (2) Soluble Protein Hydrolysate (SPH) without colitis; (3) Colitis; (4) Colitis +Control peptide (CP); (5) Colitis + SPH. A group SPH without colitis is necessary in order to know whether SPH is causing negative effects in the colon and other organs of the body.

Response 2: We are grateful for the insightful comment. In the pilot experiments, we actually set up 6 groups including 1) Control, 2) Control peptide only (CP), 3) SPH only, 4) Colitis only, 5) Colitis+CP and Colitis+SPH, but we found out that SPH and CP had no obvious negative effects on the intestinal tissues or body weight of these mice. In addition, each mouse needed to be kept in one cage to collect feces for biochemical injury assay, which requiring more space/cost in the animal facility, so SPH only and CP only groups were removed in the real experiments.

Point 3: Abstract, line 24. The authors should provide the form of SPH administration and time relation between TNBS and SPH administration.

Response 3: Colitis was induced by cutaneous sensitization with 1% TNBS on day -8 followed by 2.5% TNBS enema challenge on day 0. Control peptides and SPH were provided to the mice in the Colitis/CP or Colitis/SPH group respectively by drinking water at the final concentration of 2% W/V daily from day -10 to day 4.

Point 4:Abstract, lines 28-29. What does “SPH pre-treatment improved the DAI score and tissue injury” mean? Authors state should unequivocally whether administration of SPH increased or decreased DAI and tissue damage. This comment also applies to the Results and Discussion sections.

Response 4: We have changed “improved” for “decreased”.

Point 5:Abstract, line 31 and next parts of the manuscript. All abbreviations should be presented in their full name at the point where they appear for the first time. Full names of abbreviation presented in the Abstract should be repeated in the body of the manuscript at the place of the first use, as well as in Tables and Figure legends. Tables and Figures should be understandable without carefully reading the text of the manuscript. On the other hand, the number of abbreviations should be reduced. The excess of abbreviations makes the manuscript difficult to understand.

Response 5: We have revised them.

Point 6:Materials and Methods, line 82 and next pats of the manuscript. In the case of all drugs, regents, animals, assays and equipment, the authors should provide general and trade name of the reagent or equipment, manufacturer’s name, city, and country.

Response 6: We have replenished some relative information.

Point 7:Materials and Methods, line 87. Since 1975 is not available on the market.

Response 7: Do you mean TNBS? We think it is the most possible reagent you mentioned as we didn’t find any reagent in line 87 “shown in Figure 1A, mice were randomly divided into four groups: Control, Colitis” maybe because of wrong line. TNBS (cat# 92822) is purchased from Sigma-Aldrich Corp., St. Louis, MO, USA.

Point 8:Materials and Methods, line 85. The authors stated That “Mice had ad libitum access to water and food”. Were the animals fasted prior to the administration of TNBS enema?

Response 8: The animals were not fasted prior to the administration of TNBS enema.

Point 9:Material and Methods. How did the authors prepare 2.5% TNBS solution? The same vehicle should be used in control and SPH without colitis groups. All these data should be presented in the manuscript.

Response 9: We have replenished and presented the relative description in the methods.

The experimental colitis model was generated through 1% TNBS pre-sensitization and 2.5% TNBS enema stages. Freshly mixed acetone and olive oil in a 4:1 volume ratio by vortexing rigorously. Mixed 4 volume of acetone/olive oil with 1 volume of 5% (wt/vol) TNBS solution in H2O to obtain 1% (wt/vol) TNBS for presensitization, and mixed 1 volume of 5% (wt/vol) TNBS solution with 1 volume of 70% ethanol to obtain 2.5% (wt/vol) TNBS for enema. The above prepared solution containing TNBS was replaced by the corresponding solution containing Phosphate Balanced Saline (PBS) in the Control group.   

Point 10:Material and Methods, line 90. The authors should in detail present haw the animals were euthanized.

Response 10:  We have replenished in the manuscript as follow: “Mice were sacrificed in a CO2-euthanization station chamber equipped with an automatic air flow-regulator that is periodically inspected by the Institutional Animal Care and use committee (IACUC) at the university. ”

Point 11:Material and Methods, line 92. The authors should present more details on control peptides such as their biological source, characteristics, average molecular weight, and form provided by the manufacturer. The same information should be provided for SPH. In addition, in the case of SPH, the authors should declare whether SPH contained only peptides and proteins, or it also contained lipid admixture.

Response 11: Control peptides and SPH (both provided by Hofseth BioCare AS company, Norway) have a similar average molecular weight profile and nutritional calories. The control peptide powder is a dry, water-soluble powder provided by vital protein collagen of bovine source, while SPH is a dry, water-soluble powder which is sourced from salmon. Both powders contain greater than 98% protein and less than 0.5% lipids. Control peptides and SPH were added into drinking water at the final concentration of 2% W/V and provided to the mice in the Colitis/CP or Colitis/SPH group respectively daily from day -10 to day 4.

Point 12:Material and Methods, lines 95-97. Authors should provide the protocol approval number, as well as the date of its issue and the expiry date of its validity.

Response 12: All experiments were conducted under a protocol approved by Stanford University School of Medicine Institutional Animal Care and use committee (IACUC) in accordance with NIH guidelines (Approval code: A3213-01. Approval date: 02/19/2020 (Through 01/20/2023)).

Point 13:Figure 1. Figure 1A concerns study design and for this reason should be presented in Materials and Methods. However, Figure 1B-D presents result of the study and for this reason should be presented as a separate Figure in the section Results.

 Response 13: Figure 1 has been rebuilt according to your comment.

Point 14:Figure legends. The authors should provide the number of observations in each experimental group and state how the results are expressed in the bars.

 Response 14: We have added relative description in methods and figure legends, such as “31 mice were randomly divided into 4 groups: Control (no colitis, n=7), Colitis (n=8), Colitis/CP (with control peptide treatment, n=8) , and Colitis/SPH (with SPH treatment, n=8) ” in methods to provide the number of observations in each experimental group. Non-normally distributed data were presented with medians and quartiles (Q1:Q3) as shown in Figure 5B, 7A, 7C and quantitative data that fit a normal distribution are presented as means ± SD as shown in other figures. These results have been described in figure legends.

Point 15:Figure 2-6. Figure legends are not for presenting the interpretation of results, but they are for presenting what is shown in the figures.

Response 15: We have revised them.

Point 16:Discussion. The authors should present the importance of animal model of IBD and their limitations (PMID: 27206158; PMID: 34938152). These models are useful in the research concerning etiopathogenesis of IBD and new therapeutic concepts in this disease. It should be noted, however, that there are numerous differences between these models and clinical IBD. Therefore, the results obtained with these models may or may not be consistent with clinical effect.

Discussion. The authors presented some mechanism involved in the protective/therapeutic effect of SPH in TNBS-induced colitis. However, the authors should state that the primary mechanism of protective effect of SPH is unknown. Proteins in the digestive tract are digest into free amino acids, dipeptides, and tripeptides, and in this form are absorbed, which takes place mainly in the small intestine. Thus, there is not direct effect of SPH on the mucosa in the colon. The authors should present the concept that the protective effect of SHP is probably due to the presence of amino acids and short-chain peptides in SPH. Food is essential factor in the regulation of the renewal of gastrointestinal mucosa. Food intake stimulates cell proliferation and the growth and renewal of gastrointestinal mucosa. In the case of fasting, the renewal of gastrointestinal mucosa is inhibited leading to its atrophy (PMID: 8527816; PMID: 3710060), and the degree of weight loss in the digestive tract organs is greater than the body weight loss (PMID: 5659346). Also, in the case of parenteral nutrition, when there is no food in the gastrointestinal tract, the cell renewal of the gastrointestinal mucosa is inhibited, and hypoplasia develops (PMID: 4214726). On the other hand, refeeding leads to a rapid increase in cell proliferation in the gastrointestinal tract and the restoration of its structure (PMID: 1664265; PMID: 6142558). The mechanisms stimulating the renewal of the mucosa after food intake are mainly based the reflex of the enteric nervous system and release of gastrointestinal hormones. Among the gastrointestinal hormones that stimulate the growth of the colon, gastrin plays a special role (PMID: 25716961). Gastrin is produced and released by G cell in the stomach. The presence of small peptides and aromatic amino acids in the stomach is a main factor that acts directly on the G cells and stimulates the release of gastrin (PMID: 6806140; PMID: 25716961). Digestion of proteins in the digestive tract begins within the stomach with participation of pepsin. However, it should be stated that the essential digestion of proteins take place in the small intestine, after leaving the stomach, with the participation of pancreatic and intestinal peptidases. Moreover, pepsin is an endopeptidase that digests internal peptide bonds (PMID: 20522896). Therefore, the release of amino acids and short peptides in the stomach from food is limited. On the other hand, SPH is a product of enzymatic protein hydrolysis and most likely contains a high concentration of amino acids and short peptides, which leads to a significant release of gastrin, thereby significant stimulating the renewal of colonic mucosa and increasing the mucus production in this intestine. This concept is consistent with previous observations that also ghrelin, another factor that stimulates the growth of gastrointestinal mucosa, exhibits a protective and therapeutic effect in experimental colitis evoked by TNBS (PMID: 16697735; PMID: 19617644), dextran sulfate sodium (DSS) (PMID: 23549326; PMID: 25634696), as well as acetic acid (PMID: 26769837; PMID: 28538694). After presenting this concept with appropriate references, the authors should state this hypothesis should be verified in future research.

Response 16: Thank you very much for your insightful comment and we add some discussion as follows:

This is the first study about the protective role of SPH in IBD mice model. However, the primary mechanism of protective effect of SPH is unknown. As we know, proteins are digested into free amino acids, dipeptides, and tripeptides in the digestive tract, and are absorbed in this form mainly in the small intestine. Food is essential factor in the regulation of the renewal of gastrointestinal mucosa. Food intake can stimulate cell proliferation and the growth and renewal of gastrointestinal mucosa. In the case of fasting or parenteral nutrition, the renewal of gastrointestinal mucosa is inhibited leading to its atrophy or hypoplasia[50,51], while refeeding leads to a rapid increase of cell proliferation in the gastrointestinal tract and the restoration of its structure[52,53] due to the reflex of the enteric nervous system and the release of gastrointestinal hormones that stimulate the growth of the colon. In which, gastrin produced and released by G cell in the stomach plays a special role[54]. The presence of small peptides and aromatic amino acids in the stomach is a main factor that acts directly on the G cells and stimulates the release of gastrin[54,55]. Although digestion of proteins in the digestive tract begins within the stomach with participation of pepsin, the essential digestion of proteins take place in the small intestine with the participation of pancreatic and intestinal peptidases after leaving the stomach. Moreover, pepsin is an endopeptidase that digests internal peptide bonds[56]. Therefore, the release of amino acids and short peptides in the stomach from food is limited. SPH is a product of enzymatic protein hydrolysis and contains a high concentration of amino acids and short peptides which may lead to a significant release of gastrin stimulating the renewal of colonic mucosa, increasing the mucus production in the intestine, and improving the barrier function of the intestinal mucosa. This concept is consistent with previous observations that also ghrelin, another factor that stimulates the growth of gastrointestinal mucosa, exhibits a protective and therapeutic effect in experimental colitis evoked by TNBS[57,58], dextran sulfate sodium (DSS)[59,60], as well as acetic acid[61,62].

Above is our hypothesis about the primary mechanism of SPH that should be verified in future research. There are also other limitations exsiting in this study, such as the difference in the phenotype and function of tissue macrophages between the groups, and the protein expression levels and activity of antioxidant HO-1, SOD, NQO-1 and FTH1. In the future, we can conduct in-depth research from these aspects. In addition, although the animal models are useful in the research concerning etiopathogenesis of IBD and new therapeutic concepts in this disease, there are numerous differences between animal models and clinical IBD. Therefore, the results obtained with animal models may or may not be consistent with clinical effect and need to be verified in clinical study. 

Reviewer 3 Report

Review for the manuscript “Soluble Protein Hydrolysate ameliorates gastrointestinal inflammation and injury in TNBS-induced mouse colitis model”

Dear Editor and authors,
Thank you for the opportunity to review this interesting manuscript.

Inflammatory Bowel Diseases include inflammatory conditions (Ulcerative Colitis and Crohn's Disease) that have serious repercussions on patients' quality of life. There are numerous episodes of flares during a lifetime, even though the patient is being treated. Associated with this, the treatment is usually expensive, the patient does not always respond well to the intervention and may suffer from serious adverse effects. In this way, finding new therapies is a bright light at the end of the tunnel for patients.

Although this is an interesting manuscript, before it can be accepted for publication, I suggest some modifications.  Please, fin below my suggestions.

ABSTRACT

            In  lines 12-12, we can find “Inflammatory bowel disease (IBD) is a chronic, recurring gastrointestinal disease that can severely impair both physical health and quality of life.” I suggest that the authors use the plural for Inflammatory Bowel Diseases as there are two main entities that make up this inflammatory process.

            It is necessary to define the acronyms. For example TNF-α, IL, TGF-β1, FTH1, HMOX1, NQO-1, SOD…

INTRODUCTION

            The same suggestion I performed in the abstract applies for this section. Please, change “Inflammatory bowel disease (IBD) affects more than 1.5 million Americans with over 42 70,000 new cases diagnosed annually” for “Inflammatory bowel diseases (IBD) affect more than 1.5 million Americans with over 42 70,000 new cases diagnosed annually”.

            In  lines 42-45, we can find “Inflammatory bowel disease (IBD) affects more than 1.5 million Americans with over 42 70,000 new cases diagnosed annually (https://www.crohnscolitisfoundation.org/ sites/de- 43 fault/files/2019-02/Updated%20IBD%20Factbook.pdf).” I do not see a reason to have a link ending this sentence. Is there another way to cite this reference?

Are there other studies that used SPH in IBD models? If yes, please include it in this section. If not, before the objectives include a sentence saying that this is the first study before line 76.

METHODS

            In lines 95-87 we can read “All experiments were conducted under a protocol 95 approved by Stanford University School of Medicine Institutional Animal Care and in 96 accordance with NIH guidelines.” Please, include the date of the Ethics Committee approval and the number of this approval.

Figure 1 appears in the Methods section, but it seems to belong to the Results section. Please check. Moreover, it is on page 3, and it is first mentioned on page 5 (line 179). In fact, Figure 1 includes Methodology and results, which do not look good to the reader. Please do not mix sections in the same Figure. I suggest redoing it separately.

Is bolding necessary for citing figures in the middle of the text?

RESULTS

            As commented before, I suggest re-building Figure 1. Live Methodology as Figure 1, Start showing the results in Figure 2, and so on.
            I also suggest reducing the font size in the information windows over each photo.

DISCUSSION

In  lines 263-265 we can find “The present study demonstrates that soluble protein hydrolysate (SPH) provides robust and effective protection in a pre-clinical model of IBD, the previously described TNBS-induced rodent colitis model”. Please remove the definition of SPH because it is already in the Introduction section. Moreover, I suggest re-formulation of this sentence. Are references 4 and 19 appropriate here? Please, reformulate for a better understanding of the authors' thoughts.

In line 283 we can find “…dendritic cells (DC)…”. Dendritic cells are only mentioned here, so there is no need to use “DC”.

In  line 297 we can find “…the most abundant mononuclear phagocytes in the intestine., These cells play a critical…”. Please correct for “the most abundant mononuclear phagocytes in the intestine. These cells play a critical…”

In lines 296-304, the authors bring to light a Discussion comparing CD pathophysiology: “…large amounts of pro-inflammatory cytokines such as TNF-α and IL-6…”. What is the scenario in Ulcerative Colitis (UC)? As UC is the other important IBD, I suggest including a discussion here.

In  line 313 we can find “antigen presenting cell (APC)”. There is no need to use the acronym since it does not appear again in the text.

There are several acronyms that were not defined in the text. Some examples are It is necessary to define the acronyms. For example: TNF-α, NF-κB, IL-6, TGF-β1, IL-10, FTH1, HMOX1, NQO-1, ROS, HO, CAT, GPX, SOD…

Please, include a sentence or a whole paragraph showing the study's limitations.

CONCLUSION

            In the lines 288-289, please change rodents for mice.

Author Response

Responses to Reviewer 3 comments:

Dear reviewer, thank you very much for your timely and valuable comments, please find the point-by-point responses as follows:

Point 1: ABSTRACT

            In  lines 12-12, we can find “Inflammatory bowel disease (IBD) is a chronic, recurring gastrointestinal disease that can severely impair both physical health and quality of life.” I suggest that the authors use the plural for Inflammatory Bowel Diseases as there are two main entities that make up this inflammatory process.

            It is necessary to define the acronyms. For example TNF-α, IL, TGF-β1, FTH1, HMOX1, NQO-1, SOD…

Response 1: We found that some articles use the singular while some use the plural form for IBD. We have changed all you mentioned that need to be changed according to your comments.

We have added the definitions or full names of the acronyms when they first appeared in the article.

Point 2: INTRODUCTION

            The same suggestion I performed in the abstract applies for this section. Please, change “Inflammatory bowel disease (IBD) affects more than 1.5 million Americans with over 42 70,000 new cases diagnosed annually” for “Inflammatory bowel diseases (IBD) affect more than 1.5 million Americans with over 42 70,000 new cases diagnosed annually”.

            In  lines 42-45, we can find “Inflammatory bowel disease (IBD) affects more than 1.5 million Americans with over 42 70,000 new cases diagnosed annually (https://www.crohnscolitisfoundation.org/ sites/de- 43 fault/files/2019-02/Updated%20IBD%20Factbook.pdf).” I do not see a reason to have a link ending this sentence. Is there another way to cite this reference?

Are there other studies that used SPH in IBD models? If yes, please include it in this section. If not, before the objectives include a sentence saying that this is the first study before line 76.

Response 2: We have changed all you mentioned that need to be changed for the plural form according to your comments.

Maybe we can put the link in the references, but we think it seems more appropriate to be cited as a link in the text.

We searched the articles related to “SPH ( or Soluble Protein Hydrolysate or Protein Hydrolysate) and IBD” on pubmed and didn’t found any relative article or just found other Protein Hydrolysate from foxtail millet but not from salmon. So this is the first study about SPH from salmon in IBD model. We have added some descriptions about it and the limitation of this study in the last two paragraphs of the article.

Point 3: METHODS

            In lines 95-87 we can read “All experiments were conducted under a protocol 95 approved by Stanford University School of Medicine Institutional Animal Care and in 96 accordance with NIH guidelines.” Please, include the date of the Ethics Committee approval and the number of this approval.

 Response 3: All experiments were conducted under a protocol approved by Stanford University School of Medicine Institutional Animal Care and use committee (IACUC) in accordance with NIH guidelines (Approval code: A3213-01. Approval date: 02/19/2020 (Through 01/20/2023)).

Figure 1 appears in the Methods section, but it seems to belong to the Results section. Please check. Moreover, it is on page 3, and it is first mentioned on page 5 (line 179). In fact, Figure 1 includes Methodology and results, which do not look good to the reader. Please do not mix sections in the same Figure. I suggest redoing it separately.

 Response 3: Figure 1 has been rebuilt.

Is bolding necessary for citing figures in the middle of the text?

 Response 3: We agree with you and think it is not necessary, and have revised them.

Point 4: RESULTS

            As commented before, I suggest re-building Figure 1. Live Methodology as Figure 1, Start showing the results in Figure 2, and so on.

            I also suggest reducing the font size in the information windows over each photo.

 Response 4: Figures have been rebuilt and revised.

The font size in each photo is to be consistent with the text and can be reduced as the photos shrinks if needed.

Point 5: DISCUSSION

In  lines 263-265 we can find “The present study demonstrates that soluble protein hydrolysate (SPH) provides robust and effective protection in a pre-clinical model of IBD, the previously described TNBS-induced rodent colitis model”. Please remove the definition of SPH because it is already in the Introduction section. Moreover, I suggest re-formulation of this sentence. Are references 4 and 19 appropriate here? Please, reformulate for a better understanding of the authors' thoughts. 

In line 283 we can find “…dendritic cells (DC)…”. Dendritic cells are only mentioned here, so there is no need to use “DC”.

In  line 297 we can find “…the most abundant mononuclear phagocytes in the intestine., These cells play a critical…”. Please correct for “the most abundant mononuclear phagocytes in the intestine. These cells play a critical…”

In lines 296-304, the authors bring to light a Discussion comparing CD pathophysiology: “…large amounts of pro-inflammatory cytokines such as TNF-α and IL-6…”. What is the scenario in Ulcerative Colitis (UC)? As UC is the other important IBD, I suggest including a discussion here.

In  line 313 we can find “antigen presenting cell (APC)”. There is no need to use the acronym since it does not appear again in the text.

There are several acronyms that were not defined in the text. Some examples are It is necessary to define the acronyms. For example: TNF-α, NF-κB, IL-6, TGF-β1, IL-10, FTH1, HMOX1, NQO-1, ROS, HO, CAT, GPX, SOD…

Please, include a sentence or a whole paragraph showing the study's limitations.

Response 5: The sentence you mentioned was changed for “The present study demonstrates that SPH provides robust and effective protection in a pre-clinical TNBS-induced colitis model of IBD” , and reference 4 was deleted and 19 was adjusted to the introduction.

The discussion about the role of TNF-α and IL-6 in UC was added as follows: “TNF-α and IL-6 also play significant roles in UC pathogenesis, because an increase in TNF-α expression may contribute to defective mucosal barrier function in UC patients and exacerbate inflammation, while an increase in IL-6 plays a significant role in the trans-signaling process and chronic inflammation of UC[31]. ”

The study's limitations was showed in last paragraph. “There are also other limitations exsiting in this study, such as the difference in the phenotype and function of tissue macrophages between the groups, and the protein expression levels and activity of antioxidant HO-1, SOD, NQO-1 and FTH1. In the future, we can conduct in-depth research from these aspects. In addition, although the animal models are useful in the research concerning etiopathogenesis of IBD and new therapeutic concepts in this disease, there are numerous differences between animal models and clinical IBD. Therefore, the results obtained with animal models may or may not be consistent with clinical effect and need to be verified in clinical study. ”

Others have been revised according to your comments. 

Point 6: CONCLUSION 

            In the lines 288-289, please change rodents for mice.

Response 6: We have changed all “rodent” for “mice”.

Round 2

Reviewer 1 Report

I accept the revision of the manuscript, which is now acceptable for publication. Congratulations to the authors!

Author Response

Thank you very much.

Reviewer 2 Report

Manuscript # biomolecules-1871769

Title: Soluble Protein Hydrolysate ameliorates gastrointestinal inflammation and injury in TNBS-induced mouse colitis model

Authors: Wei et al. 

The new version of the manuscript is almost ready for publication. However, there are still some minor deficiencies.

  1. Previous comment 2 concerning the number of experimental groups. In the reply, authors stated that “In the pilot experiments, we actually set up 6 groups including 1) Control, 2) Control peptide only (CP), 3) SPH only, 4) Colitis only, 5) Colitis + CP and Colitis + SPH, but we found out that SPH and CP had no obvious negative effects on the intestinal tissues or body weight of these mice”. This statement should be present in the manuscript and the authors should write that for this reason two groups: Control peptide only and SPH only were omitted in the current study.
  2. Previous comment 6 recommending that, for each drug, reagent, animal, test and equipment, the authors should provide general and trade name of the reagent or equipment, manufacturer’s name, city, and country. In the reply, authors stated that “We have replenished some relative information.” However, complete information is still missing. For example, line 87, the city name; lines 107-108, the city name; line 136. The city and country name; line 145-146, CA is not the city name but state code; line 172, 184. Authors should carefully check the text and introduce appropriate changes.
  3. Previous comment 7. This comment was a bit unclear. The reviewer apologies for this. The comment concerned the name Sigma, not TNBS. The current form is acceptable.
  4. Authors’ reply to comment 8. “The animals were not fasted prior to the administration of TNBS enema.” This statement should be presented in the manuscript, as well as the authors should provide the period of fasting before enema.

Author Response

Responses to Reviewer 2 comments:

Dear reviewer, thank you so much for your meticulous comments, please find the point-by-point responses as follows: 

Point 1: Previous comment 2 concerning the number of experimental groups. In the reply, authors stated that “In the pilot experiments, we actually set up 6 groups including 1) Control, 2) Control peptide only (CP), 3) SPH only, 4) Colitis only, 5) Colitis + CP and Colitis + SPH, but we found out that SPH and CP had no obvious negative effects on the intestinal tissues or body weight of these mice”. This statement should be present in the manuscript and the authors should write that for this reason two groups: Control peptide only and SPH only were omitted in the current study.

Response 1: We have replenished relative description at the end of “1. Experimental Design for TNBS-Induced Colitis model” in the methods as follows:.

Noted: we actually set up two other groups (Colitis/CP and Colitis/SPH) in the pilot experiments, but it was found that SPH and CP had no obvious negative effects on the intestinal tissues or body weight of these mice, so they were omitted in the current study.

Point 2: Previous comment 6 recommending that, for each drug, reagent, animal, test and equipment, the authors should provide general and trade name of the reagent or equipment, manufacturer’s name, city, and country. In the reply, authors stated that “We have replenished some relative information.” However, complete information is still missing. For example, line 87, the city name; lines 107-108, the city name; line 136. The city and country name; line 145-146, CA is not the city name but state code; line 172, 184. Authors should carefully check the text and introduce appropriate changes.

Response 2: We are so sorry for our carelessness and carefully checked the text and replenished missing information as follows:   

line 87: BALB/c mice (The Jackson Laboratory, Bar Harbor, ME, USA)

lines 107-108 (now in line 108-109): Control peptides and SPH (both provided by Hofseth BioCare AS company, Ålesund, Norway)

line 136 (now in line 141-142): hemoccult (Beckman Coulter, Indianapolis, IN, USA)

line 145-146 (now in line 150-151): (OCT) compound media (Tissue-Tek®, Sakura Finetek USA, Inc., Torrance, CA, USA)

line 172 (now in line174):

Table 1. Sources and working conditions for immunohistochemical reagents[16].

Reagent

Manufacture

Catalog #

Clone#

Working Solution

 (μg/mL)

Incubation Time at Room Temperature (Minutes)

Rat anti-mouse CD4 mAb

Biolegend, San Diego, CA, USA

100402

GK1.5

2.5

60

Rat anti-mouse CD8 mAb

100702

53-6.7

2.5

60

Rat anti-mouse B220 mAb

103201

RA3-6B2

2.5

60

Rat anti-mouse CD68 mAb

137002

FA-11

2.5

60

Biotin-goat anti-rat lgG Ab

Jackson ImmunoResearch Laboratories, Inc., West Grove, PA, USA

112-065-062

N/A

4.0

30

HRP-streptavidin conjugate

016-030-084

N/A

5.0

30

AEC peroxidase substrate Kit

Vector Laboratories, Inc., Newark, CA, USA 

SK-4200

N/A

N/A

10-20

line 184 (now in line 189): the SpectraMax® i3x multi-mode microplate reader (Molecular Devices, San Jose, CA, USA)

In addition: we also added other missing information as follows:

line 199: iCycler (Bio-Rad Inc., Richmond, CA, USA)

line 206: Qiagen, Redwood City, CA, USA

line 213: GraphPad Prism software (Version 5.0, Graphpad Software Inc, San Diego, CA, USA).

Point 3: Previous comment 7. This comment was a bit unclear. The reviewer apologies for this. The comment concerned the name Sigma, not TNBS. The current form is acceptable.

Response 3: Thank you.

Point 4: Authors’ reply to comment 8. “The animals were not fasted prior to the administration of TNBS enema.” This statement should be presented in the manuscript, as well as the authors should provide the period of fasting before enema.

Response 4: we added the statement “The animals were not fasted prior to the administration of TNBS enema” in line 103.
